# Comparative Plastid Genome and Phylogenomic Analyses of *Potamogeton* Species

**DOI:** 10.3390/genes14101914

**Published:** 2023-10-07

**Authors:** KyoungSu Choi, Yong Hwang, Jeong-Ki Hong, Jong-Soo Kang

**Affiliations:** 1Plant Research Team, Animal and Plant Research Department, Nakdonggang National Institute of Biological Resources, Sangju 37242, Republic of Korea; hdragon@nnibr.re.kr (Y.H.); tomasx@nnibr.re.kr (J.-K.H.); 2Department of Agriculture, Forestry and Bioresources, Plant Genomics & Breeding Institute, Research Institute of Agriculture and Life Science, College of Agriculture & Life Sciences, Seoul National University, Seoul 08826, Republic of Korea; jongsookang@snu.ac.kr

**Keywords:** Potamogetonaceae, *Potamogeton*, Alismatales, plastid genome, phylogenomic analysis

## Abstract

Potamogetonaceae are aquatic plants divided into six genera. The largest genus in the family is *Potamogeton*, which is morphologically diverse with many hybrids and polyploids. Potamogetonaceae plastomes were conserved in genome size (155,863 bp–156,669 bp), gene contents (113 genes in total, comprising 79 protein-coding genes and 30 tRNA and 4 rRNA genes), and GC content (36.5%). However, we detected a duplication of the *trnH* gene in the IR region of the *Potamogeton crispus* and *P. maakianus* plastomes. A comparative analysis of Alismatales indicated that the plastomes of Potamogetonaceae, Cymodaceae, and Ruppiaceae have experienced a 6-kb inversion of the *rbcL-trnV* region and the *ndh* complex has been lost in the *Najas flexilis* plastome. Five divergent hotspots (*rps16-trnQ*, *atpF* intron, *rpoB-trnC*, *trnC-psbM*, and *ndhF-rpl32*) were identified among the *Potamogeton* plastomes, which will be useful for species identification. Phylogenetic analyses showed that the family Potamogetonaceae is a well-defined with 100% bootstrap support and divided into two different clades, *Potamogeton* and *Stuckenia*. Compared to the nucleotide substitution rates among Alismatales, we found neutral selection in all plastid genes of *Potamogeton* species. Our results reveal the complete plastome sequences of *Potamogeton* species, and will be helpful for taxonomic identification, the elucidation of phylogenetic relationships, and the plastome structural analysis of aquatic plants.

## 1. Introduction

Potamogetonaceae is an aquatic family comprising six genera (*Althenia*, *Groenlandia*, *Lapilaena*, *Zannichillia*, *Stuckenia*, and *Potamogeton*) and 110 species [1]. *Potamogeton* L. is used as food and a habitat for aquatic animals [2,3,4,5], and is divided into two subgenera *Potamogeton* and *Coleogetonia* [4]. However, the subgenus *Coleogetonia* was previously treated as an independent genus by Haynes [6], and was also treated as a synonym of genus *Stuckenia* Börner by Holub [7]. *Potamogeton* and *Stuckenia* differ in leaf shape, peduncle anatomy, and ploidy level [6,7]. *Potamogeton* species have highly similar morphological characteristics, such as leaves, seeds, and pollen, as well as various leaf shapes depending on the growing conditions [8,9,10]. Moreover, many *Potamogeton* species have undergone hybridization and polyploidization [3,11,12,13,14,15]. Hence, *Potamogeton* species are difficult to delimit taxonomically via morphological characteristics. Kaplan [11] reported that *Potamogeton* species comprise at least 50 hybrids worldwide. A previous molecular study using a 5S nuclear ribosomal array (5S-NTS) and plastid non-coding regions (*psbA-trnH* region and *trnL* intron) showed that *Potamogeton* was divided into two major groups: broad-leaved species and narrow-leaved species [16]. However, Iida et al. [17] indicated that the two major groups were not supported by the plastid non-coding region (*trnT-L* intron), and they proposed two alternative groups based on the shape of the submerged leaves and the anatomical features of the stem, and on linear submerged leaves and the presence of sub-epidermal bundles [17]. In general, the relationships among *Potamogeton* species are still unclear [16,17,18,19,20].

Plastids are important organelles for photosynthesis in plants, algae, and cyanobacteria [21,22]. Most plastid genomes (plastomes) have maternal inheritance and are often used in evolutionary and hybrid studies because of their unique characteristics [23,24]. The plastomes in angiosperms are highly conserved in terms of size, structure, order, and content. They usually comprise a pair of inverted repeat (IR) regions, a large single-copy (LSC) region, and a small single-copy (SSC) region. The genome size usually ranges from 150–160 kb, and comprises 113 genes, including, 79 protein-coding genes, 29 tRNA genes, and 4 rRNA genes [22,25]. However, recent studies have reported many variations, including rearrangement, inversion, repositioning, gene deletion, and IR, in the chloroplast genomes of the IR-lacking clade (IRLC) [26,27,28,29,30], Geraniaceae [31,32,33], Campanulaceae [34,35,36], and Orobanchaceae [37,38,39]. Therefore, plastomes have been widely used for phylogenomics [40,41,42], the development of molecular markers [43,44,45], and evolutionary studies [33,46,47,48].

Only two plastomes of Potamogetonaceae (*Potamogeton perfoliatus* and *Struckenia pectiatus*) have been reported so far [49,50], and Luo et al. [49] detected a 6 kb inversion including that in *rbcL*, *atpB*, *atpE*, *trnM*-CAU, and *trnV*-UAC in the *P. perfoliatus* plastome. In this study, we generated the complete plastomes of five *Potamogeton* species. We aimed to (1) characterize Potamogetonaceae plastomes, (2) identify divergent hotspot regions in the plastomes among Potamogetonaceae species, (3) compare plastomes among Alismatales species, (4) perform plastid phylogenomics within Potamogetonaceae and Alismatales, and (5) determine the nucleotide substitution rates among Alismatales.

## 2. Materials and Methods

### 2.1. Plant Materials and DNA Extraction

Fresh leaves of *Potamogeton* species were sampled from a natural population in South Korea. All voucher specimens were deposited at the Nakdonggang National Institute of Biological Resources (NNIBR). Total genomic DNA was extracted using the DNeasy Plant Mini Kit (Qiagen Inc., Valencia, CA, USA).

### 2.2. DNA Sequencing, Assembly, and Annotation

Genomic DNA was sequenced using the Illumina Truseq Nano DNA kit (Illumina, San Diego, CA, USA) in accordance with the manufacturer’s protocol. Approximately 8.0 GB of raw data were generated for each species. *P. perfoliatus*, *P. maackianus*, *P. crispus*, *P. wrightii*, and *P. distinctus* were sequenced to produce 26,702,712–40,111,878 total reads from the 150 bp paired-end sequences. The raw reads were assembled using GetOrganelle software v. 1.7.6.1. [50]. The coverge *of P. perfoliatus*, *P. maackianus*, *P. crispus*, *P. wrightii*, and *P. distinctus* was 2271X, 3381X, 2169X, 1123X, and 1012X, respectively. Coding genes and tRNA were annotated using GeSeq [51] and tRNAscan-SE v. 2.0. [52], respectively. OrganellarGenomeDRAW (OGDRAW) v. 1.3.1 [53] was used to draw circular maps of the plastomes of *Potamogeton* species. All plastomes were submitted to GenBank under the accession numbers listed in Table 1.

### 2.3. Comparative Genomics, Divergence Hotspot and Repeat Analysis

The six completed chloroplast genome sequences were aligned using MAFFT [54]. Nucleotide diversity (*Pi*) was determined using DnaSP v. 6.0 [55]. The step size was set to 200 bp and the window length to 600 bp.

Repeat sequences, such as forward, palindromic, reverse, and complement sequences, were analyzed using REPuter [56] with a Hamming distance of 3 and a minimum repeat size of 30 bp. The simple sequence repeats (SSRs) were detected using MISA [57]. SSRs with a minimum number of repetitions of 10, 5, 4, 3, 3, and 3 for mono-, di-, tri-, tetra-, penta-, and hexa-nucleotides, respectively, were detected.

### 2.4. Phylogenetic and Substitution Rate Analysis

The plastomes of 20 Alismatales species including six *Potamogeton* species (two of *P. perfoliatus*) and one outgroup (*Acorus gramineus*) were used. The 65 shared-protein coding genes were aligned using MAFFT v.7.222 [54]. A maximum likelihood (ML) tree was constructed on Geneious Prime using RAxML v. 8.2.11 [58] and the GTRGAMMA model with 1000 bootstrap replicates. The Bayesian inference (BI) method for phylogenies was implemented with MrBayes [59]. Markov chain Monte Carlo (MCMC) analysis was run for one million generations. The trees were sampled every 1000 generations and the initial 25% were discarded as burn-in. The remaining trees were used to build a majority-rule consensus tree.

The *dN* and *dS* rates were estimated for each of the 48 shared protein-coding genes (>200 bp) using CODEML in PAML v. 4.8 [60]. The phylogenetic tree generated in the previous section was used as the constraint tree for all rate comparisons. Codon frequencies were determined in PAML using the F3 × 4 model, and gapped regions were excluded with the “clean data = 1” parameter option. The transition/transversion ratio and *dN*/*dS* values were estimated using the initial values of 2.0 and 0.4, respectively.

## 3. Results

### 3.1. Genome Features of Potamogetonaceae Species

The plastomes of seven Potamogetonaceae species (six *Potamogeton* and one *Stuckenia* species) ranged from 155,863 bp (*P. crispus*) to 156,669 bp (*S. pectinata*). The seven Potamogetonaceae plastomes displayed a typical quadripartite structure, consisting of a pair of IRs (25,585–26,073 bp) separated by LSC (86,191–86,898 bp) and SSC (18,182–18,286 bp) regions (Figure 1, Table 1). The overall GC content was consistent (36.5%) in the seven Potamogetonaceae species. The seven Potamogetonaceae plastomes contained 113 genes, i.e., 79 protein-coding genes, 30 tRNA genes, and 4 rRNA genes. The IR regions of *P. wrightii*, *P. distinctus*, and *P. perfoliatus* had 17 genes (*trnN*-GUU, *trnR*-ACG, *trnA*-GAU, *trnI*-GAU, *trnV*-GAC, *trnL*-CAA, *trnM*-AUG, rrn4.5, rrn5, rrn23, rrn16, *rpl2*, *rpl23*, *rps7*, *ndhB*, *ycf1*, and *ycf2*), whereas those of *P. crispus*, *P. maackianus*, and *S. pectinatus* had 18 genes due to IR expansion, in which *trnH* gene was included in the IR region of the three plastomes.

The boundaries between the IR and single-copy (SC) regions of the six *Potamogeton* plastomes and the single *Stuckenia* plastome were compared (Figure 2). While the IRb/SSC boundary was similar in all *Potamogeton* species, the LSC/IRb boundary was located between *rps19* and *rpl2* regions in three species *(P. distincus*, *P. wirghtii*, and *P. perfoliatus*), whereas it was between *rps19* and *trnH* in the *P. crispus*, *P. maackianus*, and *S. pectinatus* plastomes. Five species (*P. distincus*, *P. wirghtii*, *P. perfoliatus*, *P. maackianus*, and *S. pectinatus*) had overlapping *ycf1* and *ndhF* genes, and the overlapped region between *ycf1* and *ndhF* ranged from 8 to 29 bp in length. The IRb/SSC boundary in *P. crispus* did not overlap the *ycf1* and *ndhF* genes. The SSC/IRa boundary was located in the *ycf1* gene in all *Potamogeton* species. The IRa/LSC boundary was located at the *trnH* gene in three species (*P. wrightii*, *P. distinctus*, and *P. perfoliatus)* and three species (*P. crispus*, *P. maackianus* and *S. pectinatus*) had *trnH* genes in the IRa region. The 6 kb inversion previously reported from *P. perfoliatus* was found in all Potamogetonaceae plastomes (Figure 1).

### 3.2. Repeat and Simple Sequence Repeat (SSR) Analysis

In total, 86, 90, 79, 75, 76, 86, and 88 SSRs were identified in the plastomes of *P. perfoliatus* (NC_029814), *P. perfoliatus*, *P. distinctus*, *P. wrightii*, *P. crispus*, *P. maakianus*, and *S. pectinata*, respectively (Figure 3A). Most SSRs were mononucleotide A/T repeats in all Potamogetonaceae plastomes. The plastomes of *P. perfoliatus*, *P. maackianus*, and *S. pectinate* had more mononucleotide repeats than did those of *P. distinctus*, *P. wrightii*, and *P. crispus*.

Four types of repeats (forward, reverse, complement, and palindromic) were found in the Potamogetonaceae plastomes (Figure 3B). The number of tandem repeats ranged from 34 (*P. maakianus*) to 38 (*P. perfoliatus*). Five *Potamogeton* species had 16 forward repeats, whereas *P. cirspus* had 17. Eighteen palindromic repeats were found in all the Potamogetonaceae plastomes. Three species (*P. crispus*, *P. maakianus*, and *S. pectinate*) had one reverse repeat, two species (*P. wrightii* and *P. distinctus*) had two reverse repeats, and *P. perfoliatus* had four reverse repeats in their plastomes. Complement repeats were found only in the *P. distinctus* plastome.

### 3.3. Divergence Regions in the Potamogetonaceae Plastomes

Whole plastomes within the family Potamogetonaceae (genera *Potamogeton* and *Stuckenia*) and within the genus *Potamogeton* were compared. The LSC and SSC regions showed a much higher variation than did the IR region. Among the Potamogetonaceae plastomes, the values ranged from 0 to 0.04286 (Figure 4a). Five regions, *rps16-trnQ*, *rpoB-trnC*, *trnC-psbM*, *ndhF-rpl32*, and the *atpF* intron showed higher nucleotide diversity (*Pi*) values than did other regions in their plastomes. Among the five *Potamogeton* plastomes, the *Pi* values ranged from 0 to 0.02156 (Figure 4b) and five regions (*rps16-trnQ*, the *atpF* intron, *petN-trnD*, *ccsA-ndhD*, and *ycf1*) were found to be divergence regions.

### 3.4. Comparative Analyses of the Plastomes among Alismatales

Twenty complete plastomes of Alismatales ranged from 154,516 bp (*Aponogeton desertorum*) to 179,007 bp (*Sagittaria lichunanensis*) (Table 1). The *S. lichunanensis* plastome had the longest LSC and a shorter SSC region compared with those of the plastomes of other species. The overall GC content was comparable, ranging from 36.8% (*S. lichuanensis*) to 38.2% (*Najas flexilis*). Most plastomes of Alismatales contained 113 genes (79 protein-coding genes, 30 tRNA genes, and 4 rRNA genes). However, the plastid NAD(P)H dehydrogenase (NDH) complex was lost in the *N. flexilis* plastome. The genome structure and gene order of the Alismatales plastomes are conserved. However, the *S. lichiuanensis* plastome revealed an inversion of the *psbK-trnS* region, and a 6 kb inversion was detected in the plastomes of *Sytingodium isoetifolium*, and *Ruppia brevipedunculata* (Figure 1 and Figure 5). Four types of IR/SC junctions were found in Alismatales (Figure 4). Most Alismatales plastomes had the type 3 junction. The LSC/IRa and LSC/IRb junctions were located in the *rps19*-*rpl2* region and the *psbA*-*rpl2* region, respectively. The SSC/IRa and SSC/IRb junctions were located in the *ψycf1* and the *ndhF-ycf1* region, respectively. 

### 3.5. Phylogenetic Analysis

Due to the numerous gene losses in the cp genomes of *Najas flexilis* (13 genes including 11 *ndh* genes, *infA*, and *psbH*) and *A. gramineus* (*accD*), in total, 65 shared protein-coding genes were used to reconstruct the phylogenetic relationships of Alismatales (Figure 6; Appendix A). The topologies obtained from the ML and BI trees were consistent. As a result, Alismatales was divided into two groups: (1) Alismataceae (*Sagittaria)* and Hydrocharitaceae (*Najas*, *Elodea*, *Blyxa*, and *Ottelia*), and (2) Aponogetonaceae (*Aponogeton*), Ruppiaceae (*Ruppia*), Cymodaceae (*Syringodium*), and Potamogetonaceae (*Stuckenia* and *Potamogeton*). Both groups were strongly supported by a 100% bootstrap value. The genus *Potamogeton* was sister to the genus *Stuckenia*. Within the clade *Potamogeton*, *P. maakianus* and *P. crispus* formed a subclade, and *P. wrightii*, *P. distinctus*, and *P. perfoliatus* formed another subclade. The monophyly of the genus *Potagometon* and their subclades was supported by 100% bootstrap supporting values.

### 3.6. Nucleotide Substitution Rate Analyses

In total, 48 shared protein-coding genes of 21 Alismatales plastomes, including thpse of 7 Potamogetonaceae were used to estimate the synonymous (*dS*) and nonsynonymous (*dN*) nucleotide substitution rates (Figure 7). The mean *dS* of the Alismatales plastomes was higher than the mean *dN*. While the *dN*/*dS* ratio of most genes was less than 1, the *dN*/*dS* ratios of the *rps15* gene in the plastomes of *Tofieldia thibetica*, *S. lichuanensis*, *N. flexilis*, *A. desertorum*, and *A. madagascariensis* were 1.663, 1.4746, 1.0597, 1.002, and 1.002, respectively, and that of *ClpP* in *N. flexilis* was 1.149. All Potamogetonaceae plastomes including *Stuckia* and *Potamogeton* showed a *dN*/*dS* ratio of less than 1. The *dS* of the Potamogetonaceae plastomes ranged from 0 to 0.77, the *dN* ranged from 0 to 0.1366, and the *dN*/*dS* ratio ranged from 0.0263 to 0.822. 

## 4. Discussion

The six *Potamogeton* plastomes measured approximately 156 kb in length and consistent gene content (79 protein-coding genes, 30 tRNA genes, and 4 rRNA genes). Luo et al. [49] found a 6 kb inversion in the *P. perfoliatus* plastome. By comparing the genome structure of the Potamogetonaceae (*Potamogeton* and *Stuckenia*) and Alismatale plastomes, we found the 6 kb inversion in all Potamogetonaceae plastomes (Figure 1 and Figure 6). The 6 kb inversion was also detected in the *Syringodium* (Cymodaceae) and *Ruppia* (Ruppiaceae) plastomes (Figure 5). Previous studies [61,62,63,64] suggested that inversion is likely caused by the intramolecular recombination of the repeats, and Fullerton et al. [65] suggested that G + C content affects the plastome structure. However, we did not find any evidence that the repeats or G + C content were associated with the 6 kb inversion in the Alismatales plastomes (Figure 2, Table 1).

IR expansion and construction in plastomes have been reported from diverse angiosperm lineages, such as Passifloraceae, Fabiaceae, Geraniaceae, Campanulaceae, and Poaceae [31,35,66,67,68]. The IR/SC junctions of the Potamogetonaceae plastomes can be divided into two types: (1) *trnH* in the LSC/IR junction type and (2) *trnH* in the IR region type (Figure 2 and Figure 4). *trnH* duplication was previously reported in Elaeagnaceae [69] and monocots [70], and it was hypothesized to have IR expansion [70]. First, double-strand break (DSB) events occur within the IR regions, and then the free 3′ end of the broken strand is repaired against the homologous sequence in the IR regions. We speculated that the IRb region was expanded to the *trnH* gene, which was duplicated in the IRa region via a copy correction mechanism. The *S. lichuanensis* plastome showed different IR expansions, with DSB events occurring within the IRa region and an expansion of the IRa region to the SSC (*ndhH*) region. Subsequently, a duplication of the *ndhH* gene in the newly repaired IRb was achieved.

Previous studies have conducted phylogenetic analyses of the Alismatales [49,71,72]. We reconstructed a phylogenetic tree of 21 taxa in Alismatales based on 48 shared protein-coding gene sequences. Our results showed that Alismatales was divided into two groups: (1) the petaloid clade (Hydrocharitaceae and Alismataceae) and (2) the tepaloid clade (Potamogetonaceae, Cymodaceae, Ruppiaceae, and Aponogetonaceae). This result was consistent with the results from previous studies [71,72]. 

Previous studies have suggested that the genus *Stuckenia* should be distinguished from the genus *Potamogeton* [6,16,73], whereas Wiegleb and Kaplan [11] did not support this. Our study revealed that the genus *Potamogeton* is monophyletic and sister to *Stuckenia pectinate* (Figure 6). This result also supports the taxonomic treatment of two independent genera, *Stuckenia* and *Potamogeton*. The phylogenetic relationship of *Potamogeton* was not resolved in previous studies [16,17,20,74]. All molecular phylogenetic studies of *Potamogeton* used a few plastid genes, including *psbA-trnH*, *trnT-L*, the *trnL* intron, *trnL-trnF*, *rbcL*, and nrDNA ITS regions, resulting in uncertain species delimitations of *Potamogeton* species. For example, Iida et al. [17] showed that the genus *Potamogeton* could be divided into two groups. However, they failed to distinguish between *Potamogeton gramineus* and *P. perfoliatus*, because these two species formed a clade together. Moreover, Aykurt et al. [20] suggested a sister relationship between *P. perfoliatus* and *P. nodosus*, whereas *P. perfoliatus* formed a clade with *P. richardsonii* and the clade was sister to the clade of *P. wrightii*, *P. distinctus*, *P. illinoensis*, and *P. nodosus* [74]. Our study showed that *P. perfoliatus* was sister to the clade of *P. wrightii* and *P. distinctus* (Figure 6). Due to the insertion in the *trnL*-*trnF* region, *P. crispus* was distant from *P. maackianus*, but these two species were shown to have a sister relationship in the phylogeny by Ito et al. [74]. In this study, the sister relationship between *P. crispus* and *P. maackianus* was reconstructed and supported the findings of the previous study [74]. These differences may have been caused by the misidentification of the species owing to the similar morphological characteristics and hybridization of the species. Alternatively, they may have been caused by the insufficient molecular data for the phylogenetic reconstruction of the genus *Potamogeton*. Our study suggested five regions for the phylogenetic reconstruction and species identification of the genus *Potamogeton*. The five regions, *rps16-trnQ*, the *atpF* intron, *petN-trnD*, *ccsA-ndhD*, and *ycf1* have not been used for *Potamogeton* so far, but the five regions will be used as valuable resources for determining the taxonomy and phylogenetics of the genus *Potamogeton*.

Synonymous and nonsynonymous rates provide evidence to understand the evolutionary forces in a gene [75]. The *dN*/*dS* ratio indicates the selection pressures. If the *dN*/*dS* ratio is higher than 1, the gene is under a positive selection, whereas if the ratio is less than 1, the gene is under a purifying selection [76]. We found two genes, *rps15* (*T. thibetica*, *S. lichuanensis*, *N. flexilis*, *A. desertorum*, and *Aponogeton madagascariensis*) and *clpP* (*N. flexilis*), which are under positive selection in the Alismatales plastomes (Figure 7). It has been reported that the substitution rates of the gene in the IR regions were relatively lower than those in the two SC (LSC and SSC) regions. However, the genes in the expanded IR regions did not show any reduction in substitution rates [33,47,77]. Our study also showed that the substitution rates of the *rps15* gene, which was relocated from the SSC to the IR regions in the *S. lichuanensis* plastome, were higher than those of the genes in the IR region (Figure 7). The *Potamogeton* plastomes exhibited a purifying selection (*dN*/*dS* < 1) for all genes.

## 5. Conclusions

Our study provides five newly assembled plastomes of the *Potamogeton* species. Comparative genomics of the *Potamogeton* plastomes showed that their genomes were conserved in genome size (155,863 bp–156,488 bp) and GC content (36.5%). However, IR boundary variation, such as *trnH* duplication, was detected in the *P. crispus* and *P. maakianus* plastomes. Five regions (*rps16-trnQ*, the *atpF* intron, *rpoB-trnC*, *trnC-psbM*, and *ndhF-rpl32*) were identified for phylogenetic and taxonomic studies of *Potamogeton*. Comparative genomics of the Alismatales plastomes showed that the Potamogetonaceae, Cymodaceae, and Ruppiaceae plastomes had the 6 kb inversion, and that the *trnH* duplication had occurred in the IR region of the *Stuckenia pectinate* plastome. Our phylogenomic studies using 48 shared protein-coding genes showed that Potamogetonaceae (*Potamogeton* and *Stuckenia*) was monophyletic. The synonymous and non-synonymous rates showed that the genes of the *Potamogeteon* plastomes were under purifying selection (*dN*/*dS* < 1).

## Figures and Tables

**Figure 1 genes-14-01914-f001:**
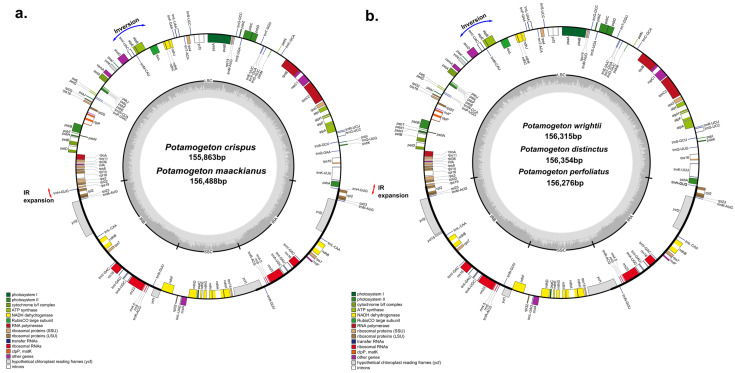
Gene map of *Potamogeton* plastid genomes. (**a**) Plastid genomes of *P. crispus* and *P. maackianus*. (**b**) Plastid genomes of *P. wrightii*, *P. distinctus* and *P. perfoliatus*. Genes drawn inside the circle are transcribed clockwise, and those outside are transcribed counterclockwise. The darker gray in the inner circle corresponds to GC contents.

**Figure 2 genes-14-01914-f002:**
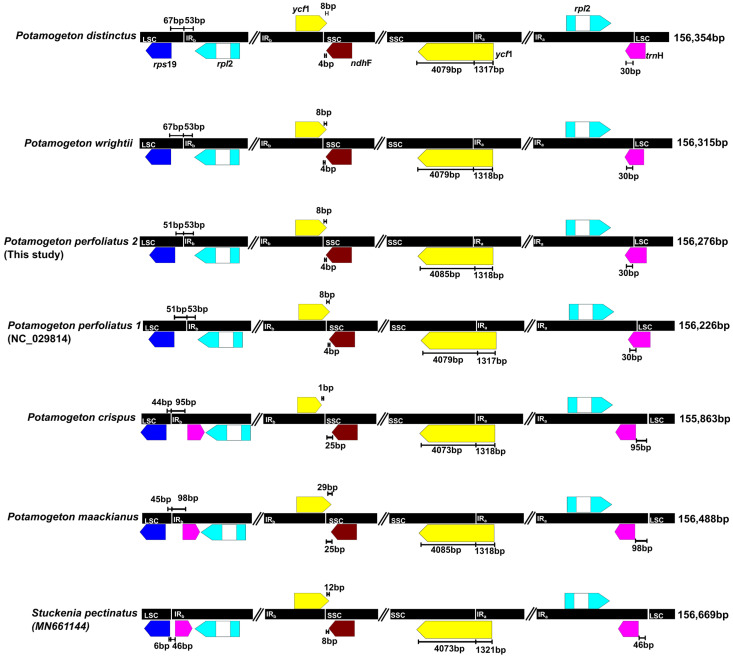
Comparison of junction between large single-copy (LSC), small single-copy (SSC), and IR regions among seven Potamogetonaceae plastomes.

**Figure 3 genes-14-01914-f003:**
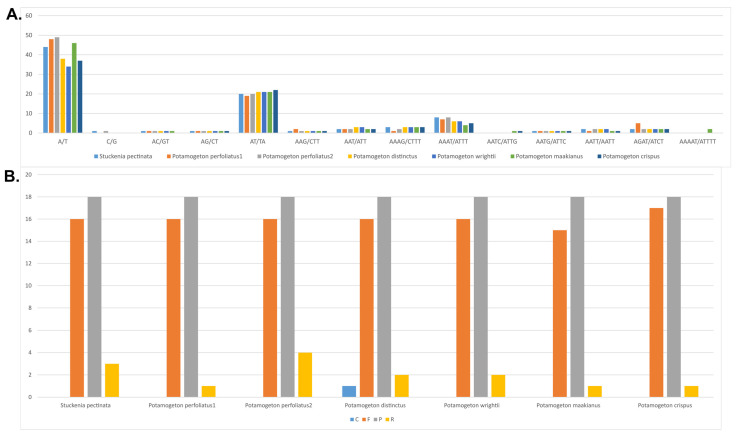
Analyses of repeat sequences in seven Potamogetonaceae plastomes. (**A**) Frequency of SSRs (simple sequence repeats). (**B**) Frequency of repeat sequences. C, complementary repeats; F, forward repeats; P, palindromic repeats; R, reverse repeats.

**Figure 4 genes-14-01914-f004:**
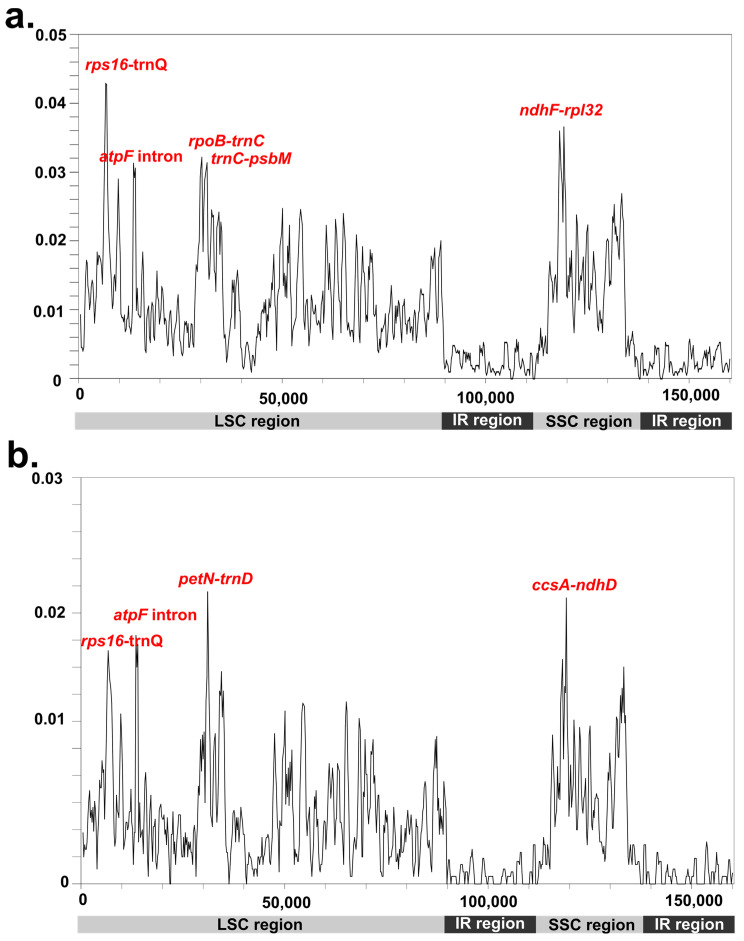
Comparison of the nucleotide variability (*Pi*) values (**a**) compared among Potamogetonaceae species and (**b**) compared among *Potamogeton* species.

**Figure 5 genes-14-01914-f005:**
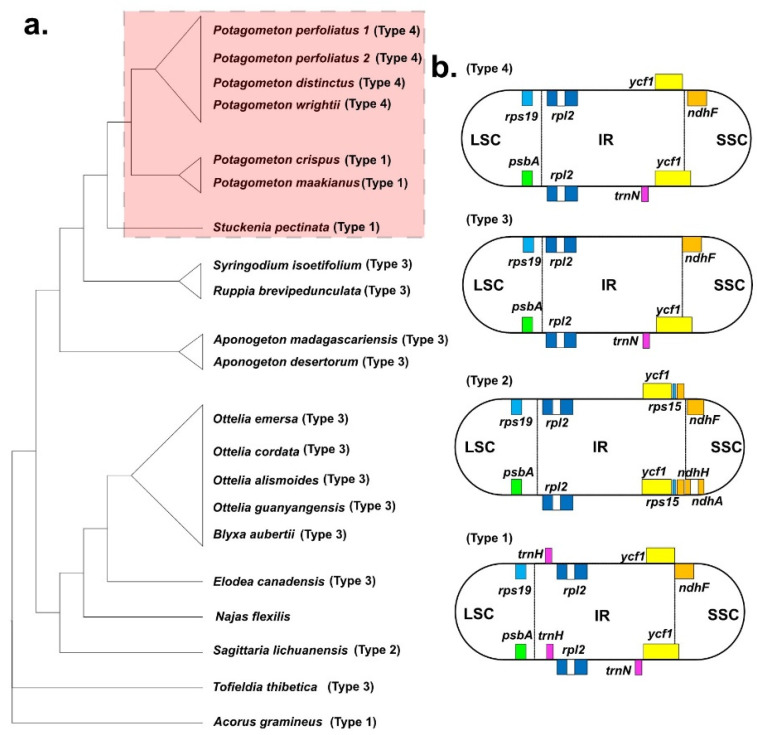
(**a**) The phylogenetic trees were constructed based on 65 coding genes of 21 Alismatales plastomes. (**b**) Types of junction between large single-copy (LSC), small single-copy (SSC), and IR regions in the Alismatales plastomes.

**Figure 6 genes-14-01914-f006:**
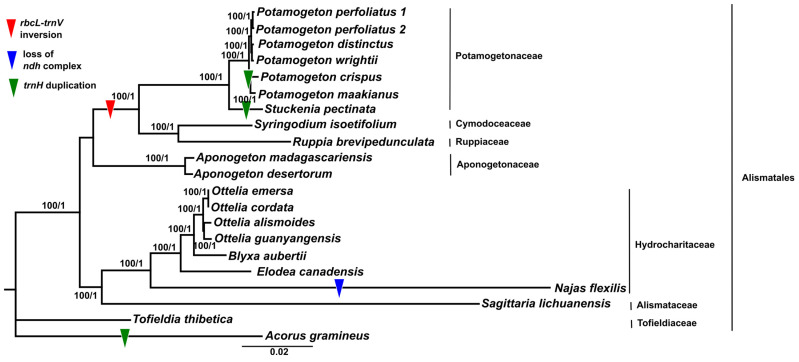
Phylogenetic tree constructed using the maximum likelihood (ML) and Baysian inference (BI) methods based on 65 plastid protein-coding genes. The number above the lines indicates bootstrap values/BI posterior probabilities.

**Figure 7 genes-14-01914-f007:**
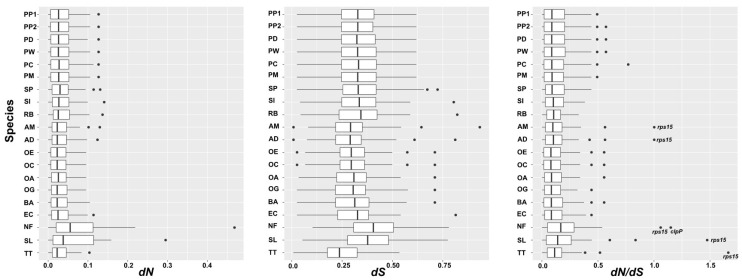
Nonsynonymous (*dN*) and synonymous (*dS*) substitution rates and *dN*/*dS* values of 48 plastid protein-coding genes across Alismatales. PP1, *Potamogeton perfoliatus1*; PP2, *Potamogeton perfoliatus 2*; PD, *Potamogeton distinctus*; PW, *Potamogeton wrightii*; PC, *Potamogeton crispus*; PM, *Potamogeton maackianus*; SP, *Stuckenia pectinate*; SI, *Syringodium isoetifolium*; RB, *Ruppia brevipdeunculata*; AM, *Aponogeton madagascariensis*; AD, *Aponogeton desertorum*; OE, *Ottelia emersa*; OC, *Ottelia cordata*; OA, *Ottelia alismoides*; OG, *Ottelia guanyangensis*; BA, *Blyxa aubertii*; EC, *Eldea canadensis*; NF, *Najas flexilis*; SL, *Sagittaria lichuanensis*; TT, *Tofieldia thibetica*.

**Table 1 genes-14-01914-t001:** Comparison of the plastid genome features of *Potagometon* species.

Order	Family	Species	Length (bp)				GC Contents	NCBI Accession Number
			Total	LSC	SSC	IR		
Acorales	Acoraceae	*Acorus gramineus*	152,849	82,977	18,228	25,822	38.7%	NC_026299
Alismatales	Tofieldiaceae	*Tofieldia thibetica*	155,512	84,584	18,151	26,388	37.4%	NC_029813
	Alismataceae	*Sagittaria lichuanensis*	179,007	99,125	13,278	33,302	36.8%	NC_029815
	Hydrocharitaceae	*Najas flexilis*	156,366	88,697	15,266	31,201	38.2%	NC_021936
		*Eldea canadensis*	156,700	86,194	17,808	26,349	37.0%	NC_018541
		*Blyxa aubertii*	158,187	87,799	18,804	25,792	36.5%	MK940507
		*Ottelia guanyangensis*	157,362	87,230	19,004	25,564	36.7%	MK940522
		*Ottelia alismoides*	157,880	87,699	19,067	25,557	36.6%	MK940517
		*Ottelia cordata*	157,896	87,665	19,121	25,555	36.6%	MK940519
		*Ottelia emersa*	157,896	87,665	19,121	25,555	36.6%	MK940520
	Aponogetonaceae	*Aponogeton desertorum*	154,516	85,760	19,890	24,433	36.9%	MK570533
		*Aponogeton madagascariensis*	155,669	86,896	19,869	24,452	36.9%	MK570534
	Cymodoceaceae	*Ruppia brevipedunculata*	158,943	88,857	19,130	25,478	35.8%	NC_051974
		*Syringodium isoetifolium*	159,333	89,055	19,160	25,559	35.9%	MZ325253
	Potamogetoncaceae	*Stuckenia pectinata*	156,669	86,285	18,237	26,073	36.5%	MN661144
		*Potamogeton perfoliatus1*	156,226	86,764	18,238	25,612	36.5%	NC_029814
		*Potamogeton perfoliatus2*	156,276	86,821	18,231	25,612	36.5%	This study (OQ561452)
		*Potamogeton maackianus*	156,488	86,833	18,221	25,717	36.5%	This study (OQ561451)
		*Potamogeton crispus*	155,863	86,191	18,182	25,745	36.5%	This study (OQ561449)
		*Potamogeton wrightii*	156,315	86,827	18,282	25,603	36.5%	This study (OQ561453)
		*Potamogeton distinctus*	156,354	86,898	18,286	25,585	36.5%	This study (OQ561450)

LSC, large single-copy region; SSC, small single-copy region; IR, inverted-repeat region.

## Data Availability

All the newly sequenced sequences in this study are available from the National Center for Biotechnology Information (NCBI) (https://www.ncbi.nlm.nih.gob/ (accessed on 9 April 2023); accession numbers: OQ561449-OQ561453; see Table 1). Information for other samples used for phylogenetic analysis downloaded from GenBank can be found in Table 1.

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
