# Peer review of "Comparative Plastid Genome and Phylogenomic Analyses of Potamogeton Species"

_genes, 2023, doi:10.3390/genes14101914_

Round 1
Reviewer 1 Report
Dear Authors,
Nicely done, good work. The only missing parts are the abbreviations, please read the whole text and look for the missing abbreviations.
- overall the English of the manuscript is well written, but some misspellings and typos are present, which decrease the readability of the manuscript
- Table 1. contains abbreviations that are not mentioned before, please clarify the meaning of the abbreviations (Figures and Tables need to stand alone too)
- Figure 1: writing on the figure is small, please increase the visibility
- Previously mentioned phenomenon is stands for all the figures
- Read through the maunscript and correct all the headers, etc., based ont he given template - before all the citations a space is missing, please correct them
Fine, only a few misspellings and typos are present.
Reviewer 2 Report
The study reported the complete chloroplast genomes of five Potamogeton species. The genome structural variations and phylogenetic relationships were further analyzed. Notably, the research identified regions of structural variation and pinpoints five highly variable sites, providing crucial insights into the evolutionary dynamics of chloroplast genomes in Potamogeton plants. Furthermore, through the construction of high-resolution phylogenetic trees, the authors have mitigated the low resolution issues that have plagued prior studies, thereby offering valuable guidance for morphological classifications. However, several noteworthy issues merit attention, as outlined below.
Major comments
1. In the Abstract (Line 19-20), it is somewhat ambiguous to claim that Potamogetonaceae was monophyletic, given that only two out of seven genera were included in the analysis.
2. In the Introduction (Line 34-36), the authors mentioned the high morphological similarity between Potamogeton species in terms of leaves, seeds, and pollen. However, no morphological data is presented in the entire text. It is advisable to include some morphological comparative evidence or clarify the focus of the study.
3. In the Materials and Methods section, there is no mention of the quality control procedures applied to the raw data before assembly, nor any mention of subsequent inspection and correction of the assembly results. More detailed information about these processes should be provided.
4. In the Materials and Methods section, the authors mentioned that the plastomes of Potamogetonaceae and Alismatales were compared using mVISTA. Please specify where the results of this comparison can be found or provide more information on this analysis.
5. In the phylogenetic analysis, only the Maximum Likelihood method was used for inferring phylogenetic relationships. Please consider adding the Bayesian inference method to enhance the robustness of your phylogenetic analysis.
6. In the results, the authors mentioned that most plastomes of Alismatales contained 113 genes (79 protein-coding genes), but in the final analysis, only 48 shared protein-coding genes were used. Please clarify whether this difference is primarily due to variations between different families or if it resulted from not re-annotating sequences from NCBI. A table summarizing gene count statistics for each species should be provided.
7. In Figure 3, the organization of SSRs and repeats for interspecies comparison appears cluttered. Please consider presenting SSRs and repeats for all species in two separate figures for better clarity.
Minor comment
8. The number of genera in Potamogetonaceae is inconsistent between the Abstract (Line 11) and the Introduction (Line 28).
9. In Tab. 1, there is an inconsistency between the accession numbers and species names for Potamogeton perfoliatus 1 and Potamogeton perfoliatus 2 compared to Figure 2.
10. In Figure 5, it's not clear what the phylogenetic tree in panel (a) was constructed based on.
The language needs tightening up and editing for English sense.
Round 2
Reviewer 2 Report
In this revised version, the authors further improved the analyses. I think most of the concerns raised by reviews have been adequately addressed and the manuscript has been improved.
The writing needs improvements throughout.